# National differences in dissemination and use of open access literature

**Marc-André Simard** [1]*, **Gita Ghiasi** [1], **Philippe Mongeon**[2], **Vincent Larivière**[3]

**1** École de bibliothéconomie et des sciences de l'information, Université de Montréal, Montréal, Québec, Canada, **2** School of Information Management, Dalhousie University, Halifax, Nova Scotia, Canada, **3** École de bibliothéconomie et des sciences de l'information and Observatoire des sciences et des technologies, Université du Québec à Montréal, Montréal, Québec, Canada

* marc-andre.simard.1@umontreal.ca

## Abstract

Open Access (OA) dissemination has been gaining a lot of momentum over the last decade, thanks to the implementation of several OA policies by funders and institutions, as well as the development of several new platforms that facilitate the publication of OA content at low or no cost. Studies have shown that nearly half of the contemporary scientific literature could be available online for free. However, few studies have compared the use of OA literature across countries. This study aims to provide a global picture of OA adoption by countries, using two indicators: publications in OA and references made to articles in OA. We find that, on average, low-income countries are publishing and citing OA at the highest rate, while upper middle-income countries and higher-income countries publish and cite OA articles at below world-average rates. These results highlight national differences in OA uptake and suggest that more OA initiatives at the institutional, national, and international levels are needed to support wider adoption of open scholarship.

## Introduction

Open Access (OA) dissemination makes research outputs freely available on the public Internet, allowing users to read, download, copy, distribute, print, search, or link to the full text without any financial, legal, or technical barriers, in accordance with an open copyright licence [1]. One of the milestones of the widespread awareness of OA was 2001's Budapest Open Access Initiative (BOAI), which established the first clear distinction between the two main types of OA: self-archiving (green OA) and a brand-new generation of journals that would allow scholarly material to be distributed in its full form, freely on the publisher's website (gold OA). Gold open access journals may be associated with article processing charges, originally intended to cover publication costs. However, the origins of OA go back to the 1970s, when computer scientists were already sharing their papers with File Transfer Protocol (FTP) servers. The first OA journals appeared in the late 1980s with *New Horizons in Adult Education*, *Psycholoquy* (1989), and *Public-Access Computer systems Review* (1989). In the early 1990s, the movement had already picked up some steam with the creation of a dozen more electronic journals, but more importantly, the creation of the first mainstream self-archiving repository

**Data Availability Statement:** Restriction apply to both datasets used to generate the aggregate data. The Web of Science data is owned by Clarivate Analytics. To obtain the bibliometric data in the same manner as authors (i.e. by purchasing them),

readers can contact Clarivate Analytics at the following URL: https://clarivate.com/webofsciencegroup/solutions/web-of-science/contact-us/. Open Access data are available for free on the Unpaywall database (https://unpaywall.org/products/api). Data from those three sources were combined to create new dataset which we used to create the Z-scores and to complete the analyses for the study. The final data set with the Z-scores has been shared which can be found here: https://github.com/masim362/Global_OA_paper. Open Access data are available for free on the Unpaywall database (https://unpaywall.org/products/api). Data from those three sources were combined to create new dataset which we used to create the Z-scores and to complete the analyses for the study. The final data set with the Z-scores has been shared under a Creative Commons Attribution 4.0 International License and can be found here: https://github.com/masim362/Global_OA_paper".

**Funding:** Funding by the Canada Research Chair program (grant 950-231768) and the SSHRC Joseph Armand Bombardier Master's Scholarship. However, the funders had no role in study design, data collection and analysis, decision to publish, or preparation of the manuscript.

**Competing interests:** The authors have declared that no competing interests exist.

*arXiv* (1991) by Paul Ginsparg, which allowed physics and engineering researchers to deposit preprint version of their papers for free on the Internet. The year 1991 also marked the publication of Dr. Allen Bromley's, then advisor to President George H. W. Bush, *Policy Statements on Data Management for Global Change Research*. The statement pleaded for global open access, open data, and preservation of this data in open repositories [2]. In 2000, BioMed Central, the first for-profit OA publisher, was founded.

Nearly half a century since the first use of FTP servers to share scientific papers, the OA movement is still gaining momentum, with the implementation of several OA policies by funders and universities worldwide and the development of new business models by for-profit and not-for-profit publishers. According to several studies, nearly half of research articles could be available online at no cost [3–5]. Previous studies have analyzed the availability of articles [3–5], their citation advantage [4, 6, 7], and compliance with OA mandates [8]. It has often been argued that OA could improve global participation in science, especially for developing countries. However, few articles [9, 10] have compared how different countries, especially developing ones, use OA literature. This paper provides a contemporary portrait of the adoption of OA worldwide, distinguishing between publishing in OA and citing OA publications and between the two main types of OA (green and gold).

## Background

## Open access models

This study is based on the original Budapest Open Access Initiative (2002) OA definition, which distinguishes between the two original models of OA: Gold Open Access and Green Open Access (or self-archiving). Gold journals are specifically committed to open access and allow users to access their articles for free directly on their website. They may or may not charge article processing charges (APCs) to generate revenues. Since the early 2000s, the amount of gold OA journals in the Directory of Open Access Journals (DOAJ) has been growing exponentially, going from around 20 in 2002 to 16,621 as of July 17th, 2021. Indonesia has the highest number of OA journals with 1852, followed closely by the United Kingdom (1814) and Brazil (1637). Hybrid open access is another type of OA publishing that the scientific community has generally criticized because of "double-dipping": hybrid articles are published in subscription-based journals that offer the option of paying an additional fee, with publishers thus getting paid twice for the same article [11, 12]. The bronze OA model has been recently introduced by Piwowar et al. [4]. Bronze OA refers to publications that are freely accessible on the journal or publisher's website but do not have an identifiable open license.

Self-archiving or green open access is achieved through depositing an article in an open electronic archive or repository. These electronic archives may be thematic (i.e. arXiv, *BioRxiv*), institutional (i.e. Harvard's *DASH*), or personal (i.e. a personal website). As of November 2021, 5,764 and 4,629 repositories were listed in the Directory of Open Access Repositories (OpenDOAR) and the Registry of Open Access Repositories (ROAR), respectively. The repositories are distributed as follows according to the ROAR data: Europe (36.8%), Asia (22.3%), North America (22.1%), South America (12.8%), Africa (3.9%), and Oceania (2.1%). Laakso [13] underlined the large gap between the potential self-archiving and the actual self-archiving done by researchers: over 80% of all scientific production, including 62% of articles published by the top 100 biggest publishers, could be available in OA based on publishers' self-archiving policies. However, other studies have estimated that the current share of articles available in green OA is between 10 and 25% [4].

Several studies assessed the number and percentage of scholarly papers freely available online in different periods and disciplines. They found that 20% to 54% of research articles

were available online at no cost [3–5, 14, 15]. According to a study by Piwowar et al. [4], 7.4% of a sample of 100,000 articles indexed by the Web of Science between 2009 and 2015 were available via gold OA, while 11.5% of the same articles were available on repositories. Other types of OA accounted for 17.2% of the sample, for a total of 36.1% of articles being available in OA. Using a sample of Unpaywall's 2017 data, the same study found a proportion of 14.3% gold OA articles, 9.1% green OA articles, and 23% of articles available with other types of OA methods. One should note that in their study, Piwowar and her colleagues considered green and gold as mutually exclusive categories.

A recent paper by Robinson-Garcia et al. [16] analyzed the OA uptake by 963 universities worldwide by combining data from Web of Science and Unpaywall. Their results showed that the median share of OA publications of universities worldwide of 43%, with European universities generally leading the pack. In their analysis of the geographical distribution of publications about OA, Miguel et al. [17] showed that about 30% of a sample of 1179 articles about OA came from the United States, 13% from the United Kingdom, and 6% from Germany and Spain. In terms of OA usage (i.e., references to OA publications) by country, a 2009 study by Evans and Reimer combined extensive bibliometric data with the World Bank and the UNESCO data on per capita gross national income (GNI) and found that paywalled articles were disproportionately cited by researchers in rich countries. Inversely, they found that OA publications were disproportionately cited in developing countries, except in the very poorest countries where electronic access may be limited. They concluded that while the influence of OA is more modest than it has been proposed in the past, their results supported OA's potential to improve global participation in science. In the biomedical fields, Iyandemye and Thomas [10] found a negative correlation between the GNI and the share of OA publications of a country. This could be explained APCs waivers for low-income countries.

OA mandates and policies have often been mentioned in the scientific literature as a possible way to encourage researchers and journals to publish in OA. An OA mandate is a policy adopted by a funding agency or a research institution that requires researchers to make their publications open access [11]. Institutional mandates are generally less restrictive and tend to vary from one institution to another. Funding agency mandates are usually stricter: they impose a contractual obligation to publish in OA. Failure to comply may ultimately lead to the withholding of the funding. Gold OA mandates have been criticized for restricting the freedom of researchers by forcing them to publish in gold OA journals. In contrast, green mandates are not very restrictive since most publishers already allow researchers to deposit their papers in OA repositories [11]. The Registry of Open Access Mandates and Policy (http://roarmap.eprints.org/) indexes OA mandates and policies adopted by universities, research institutions and funding agencies. According to ROARMAP, in the last fifteen years, the number of OA mandates increased from 123 in 2005 to 1,094 in 2021. The majority of OA mandates are currently in Europe (N = 695), followed by the Americas (N = 240), Asia (N = 82), Oceania (N = 42), and Africa (N = 35). Although ROARMAP is a valuable tool to overview the state of open access policies worldwide, one must consider that information might be incomplete since it depends on voluntary registration by funding institutions.

Compliance with mandates varies based on the type of mandate (institutional or funding) and across disciplines [8, 18]. Larivière and Sugimoto [8] found a positive relationship between OA mandates and availability: out of 1.3 million articles subject to a mandate, around two-thirds were available for free on the Internet, compliance with the mandate ranging from 29% in some social sciences to 85% in biomedical research. Another study by Gargouri et al. [19] found that more publications were made available online for free (about 60%) by researchers subjected to an institutional mandate to self-archive, compared to approximately 15% of publications from researchers without a self-archiving mandate. On the other hand, Iyandemye and

Thomas [10] found no clear relationship between the number of open access policies in a country and its percentage of open access publications.

## Initiatives and platforms

Over the last 20 years, several initiatives and platforms have been developed around the world to support OA dissemination. At the world level, UNESCO created the *Global Alliance of Open Access Scholarly Communication Platforms to Democratize Knowledge* to improve access to scholarly communications following a multicultural, multilingual, and multi-themed approach [20]. UNESCO has also been working on a Recommendation on Open Science that aims to define "shared values and principles for open science" and "identify concrete measures on Open Access and Open Data" [21]. Member states officially adopted the Recommendation in November 2021 [22].

**Europe and Plan S.**   In Europe, Plan S, an ambitious project led by cOAlition S, aims to make every scientific article supported by public funds freely available on the Internet by 2021 and remove or lower the article processing charges to low-income or middle-income countries [23]. To illustrate the potential impact of Plan S on the scholarly communication infrastructure, in 2017, 35% of papers published in *Nature* and 31% of those in *Science* were funded by at least one of the coalition's members [24].

**North America.**   In Canada, the non-for-profit platform Érudit is now the largest disseminator of French papers in North America, with more than 95% of its catalogue available in OA [25]. In the United States, the PubMed Central repository was created in 2000 by the United States National Library of Medicine. In 2007, the National Institute of Health (NIH) introduced an OA policy (NIH Public Access Policy) which mandated that every article funded by the NIH had to be available for free within 12 months after publication [26], which has contributed to the expansion of PMC. In 2021, PMC indexed over 5.9 million papers, including over 8000 scientific journals. Discussions around a new OA initiative comparable to Plan S that would make every publication funded by the federal government available online for free [27] met strong opposition by publishers. In a letter to the US president signed by over 135 organizations, including the American Chemical Society, Elsevier, Wiley, and Wolters Kluwer, the Association of American Publishers argued that such a policy would negatively affect innovation and peer review in science [28].

**Latin America and the Caribbean.**   In South America, the Open Science movement is well established and supported by major initiatives such as the Scientific Electronic Library Online (SciELO). Created by the Brazilian government in 1997, SciELO aims to provide the infrastructure needed to assist the publishing industry in developing countries and to give more visibility to scientific articles [29]. In 2021, SciELO indexed over 874,000 OA articles published in 1,768 journals. Founded in 2018, AmeliCA is a publishing cooperative led by UNESCO and the Latin American Council of Social Sciences that aims to provide a sustainable and non-profit OA infrastructure for South America and the Global South. Redalyc is a digital library of open access journals hosted by Universidad Autónoma del Estado de México. It currently indexes over 1400 journals published by 669 institutions from 25 countries. Latin America also hosts the LA Referencia (https://www.lareferencia.info/en/) repository, where authors can deposit their work.

**Asia and the Pacific.**   In Asia, J-STAGE is an initiative funded by the government of Japan that aims to accelerate the dissemination of science by helping research institutions with the publication process. The J-STAGE database currently indexes over 5 million articles, including 4.7 million available in OA. In China, the Chinese Academy of Science (CAS) launched the China Open Access Journals (COAJ), which indexes OA journals from China. The CAS also

hosts the CAS IR Grid (http://www.irgrid.ac.cn/) as a collective knowledge repository where it can ensure access and preservation of its knowledge production. Since 2015, CAS has mandated its research institution to provide an OA version of their articles within 12 months of publication [30].

**Africa.** In Africa, access to the Internet remains a major problem in several countries, especially in Sub-Saharan Africa (Broadband Commission, 2019). Founded in 1998 by the International Network for the Availability of Scientific Publication, the African Journal OnLine (AJOL) project aims to improve dissemination and access to research in Africa for African researchers and researchers worldwide. To this day, AJOL indexes over 525 journals (including 263 OA journals), and over 172 articles, with nearly two-thirds available in OA. In 2009, South Africa launched SciELO SA (http://www.scielo.org.za/).

## Research objectives

This paper aims to provide a global picture of OA adoption by countries, using two indicators: publications in OA and references made to articles in OA. Our research questions are as follows:

1. How does the share of OA publications differ between countries?

2. How does the share of references to OA publications differ between countries?

3. What is the relationship between OA publishing and citing frequency within counties?

4. What is the relationship between country income level and OA adoption?

## Methods

We collected all articles and reviews in the Web of Science (WoS) published between 2015 and 2019 (N = 8,590,184). Science being increasingly collaborative, publications are generally authored by multiple individuals, often from different institutions and countries, among whom the work has typically been unevenly distributed (Larivière et al. 2015). We argue that choices in terms of cited literature and publication venues are predominantly made by authors who played a leading role in the research. Thus, when assigning a publication to a country, we only use the institutional affiliations of the first and corresponding authors. When the first and corresponding authors are affiliated with multiple countries, the publication is fully counted for all the countries. We divided countries into four income level groups based on the 2021–2022 World Bank's country classification (https://blogs.worldbank.org/opendata/new-world-bank-country-classifications-income-level-2021-2022).

We used the Unpaywall (http://unpaywall.org) database to determine the OA availability and OA type of a paper or reference. Because we use the DOI to link the WoS and Unpaywall data, the 381,101 WoS publications that do not include a DOI, and the 71,408 dois that were not found in Unpaywall were dropped, leaving us with a dataset of 8,137,675 publications. This data indicates whether the paper is published in OA or not, and the OA category it belongs to (gold, green, hybrid or bronze). Some of our analyses include all OA categories while others were limited to green and gold to stay true to the original BOAI definition. These two categories are not mutually exclusive and papers that are published in a gold OA journal and found in a repository were counted in both categories.

The percentage of OA publications varies drastically across disciplines [31] and the distribution of publications across fields varies by country. To account for this, we calculate the percentage of gold and green OA publications for each country and field, and then calculate the

field-weighted average z-score for each country. The field-weighted average z-score is calculated as follows:

$$z_{c,f} = \frac{(x_{c,f} - \mu_f)}{\sigma_f}$$

Where:

- $x$ is the share of OA publications or references of country $c$ in field $f$,

- $\mu$ is the average share of OA publications or references for all countries in field $f$, and

- $\sigma$ is the standard deviation of the share of OA publications or references for all countries in field $f$.

This makes values from the different indicators comparable and recenters them around the average ($z = 0$), negative values indicated lower than average attention and positive values indicating higher than average attention. These z-scores are used for all the country-level analyses below.

## Results

Tables 1 and 2 present an overview of our dataset by discipline and type of OA in terms of publications and references, respectively. In Table 1, we see that 42.9% of the articles published during the 2015–2019 period were Open Access at the time of our data collection. The proportion of publications in OA varies by fields, ranging from 21.2% in the Humanities to 50% in the Medical and Health Sciences. The very small proportion of papers that are in gold OA only (0.5%–6.1%) shows the large overlap between green and gold OA.

In Table 2, we see that the share of references to OA papers also varies by fields with Medical and Health Sciences citing more OA papers. However, we can observe that the proportion of references to gold OA articles (7.2%) is generally lower than the proportion of gold OA publications presented in Table 1 (18.1%). Overall, the proportion of OA references (39.6%) is lower than the proportion of OA publications (42.9%).

Figs 1 and 2 reveal that Sub-Saharan African countries tend to publish and use OA more than the rest of the world. In North America, the United States publishes and cites OA more than the world average, however, Canada publishes in OA less often but cites OA more often. South American countries are generally slightly over the average, with western countries such as Ecuador, Peru, and Chile generally publishing and citing OA more, while Brazil is at the world average. In Europe, Western European countries mostly publish and cite OA, while the

**Table 1. Proportion of OA publications by type and by fields (2015–2019).**

| Field | N | OA | Gold | Green | Gold Only | Green Only |
|---|---|---|---|---|---|---|
| Natural Sciences | 4,282,099 | 45.4 | 19.9 | 36.3 | 3.7 | 20.1 |
| Engineering and Technology | 2,307,970 | 30.4 | 13.0 | 21.4 | 4.9 | 13.3 |
| Medical and Health Sciences | 2,634,315 | 50.0 | 20.8 | 40.4 | 2.0 | 21.6 |
| Agricultural Sciences | 402,115 | 35.9 | 17.1 | 22.0 | 6.1 | 11.0 |
| Social Sciences | 795,087 | 35.5 | 7.9 | 29.8 | 1.9 | 23.8 |
| Humanities | 266,985 | 21.2 | 5.9 | 15.8 | 2.6 | 12.5 |
| Unknown | 33,666 | 35.8 | 2.2 | 31.3 | 0.5 | 29.6 |
| All Fields | 8,590,184 | 42.9 | 18.1 | 33.8 | 3.4 | 19.1 |

**Table 2. Proportion references to OA by type and by fields.**

| Field | Number of references | OA | Gold | Green | Gold Only | Green Only |
|---|---|---|---|---|---|---|
| Natural Sciences | 150,284,096 | 40.9 | 7.5 | 32.4 | 0.7 | 14.8 |
| Engineering and Technology | 66,541,412 | 25.9 | 5.0 | 20.6 | 0.8 | 11.9 |
| Medical and Health Sciences | 86,728,869 | 50.6 | 9.2 | 37.1 | 0.6 | 14.2 |
| Agricultural Sciences | 11,536,124 | 33.4 | 8.1 | 22.2 | 1.2 | 7.7 |
| Social Sciences | 21,145,993 | 32.7 | 3.3 | 28.3 | 0.2 | 20.2 |
| Humanities | 1,785,285 | 27.0 | 2.8 | 23.1 | 0.3 | 16.2 |
| Unknown | 900,121 | 39.9 | 4.9 | 36.2 | 1.1 | 27.8 |
| All Fields | 338,921,900 | 39.6 | 7.2 | 30.6 | 0.7 | 14.2 |

trend is opposite for the Eastern European countries. In Asia, North Africa and the Middle East, most countries publish or cite OA less often.

Fig 3 illustrates the relationship between the weighted and normalized OA publications and the weighted and normalized references. The Pearson correlation shows a moderate relationship between the two indicators ($r = 0.4805$; $p < 0.001$), indicating that the more a country publishes in OA, the more it is likely to make references to OA papers.

Fig 4 shows plots of countries according to their OA publications and references compared to the field-normalized world average. Overall, as Fig 1 suggests, the two indicators correlate. The correlation is, however, much stronger for low-income ($r = 0.884$; $p < 0.001$) and lower-middle-income countries ($r = 0,827$; $p < 0.001$) than for upper-middle-income countries ($r = 0.524$; $p < 0.001$) and high-income countries ($r = 0.630$; $p < 0.001$). On average, the different groups are positioned in different quadrants: the high-income countries are mostly positioned in the first and third quadrants, while upper-middle-income countries are in the third

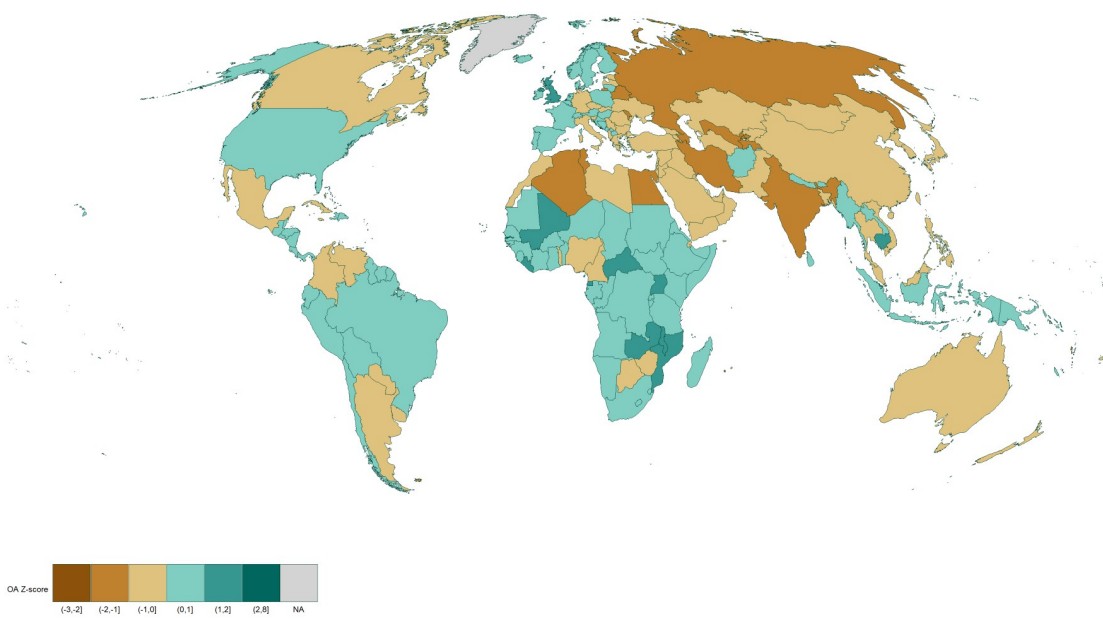

**Fig 1. Weighted z-score and normalized map of the number of OA publications by country.** Red indicates that a country is above the world average, blue indicates it is below the world average. White represents the world average. Contains information from OpenStreetMap and OpenStreetMap Foundation, which is made available under the Open Database License.

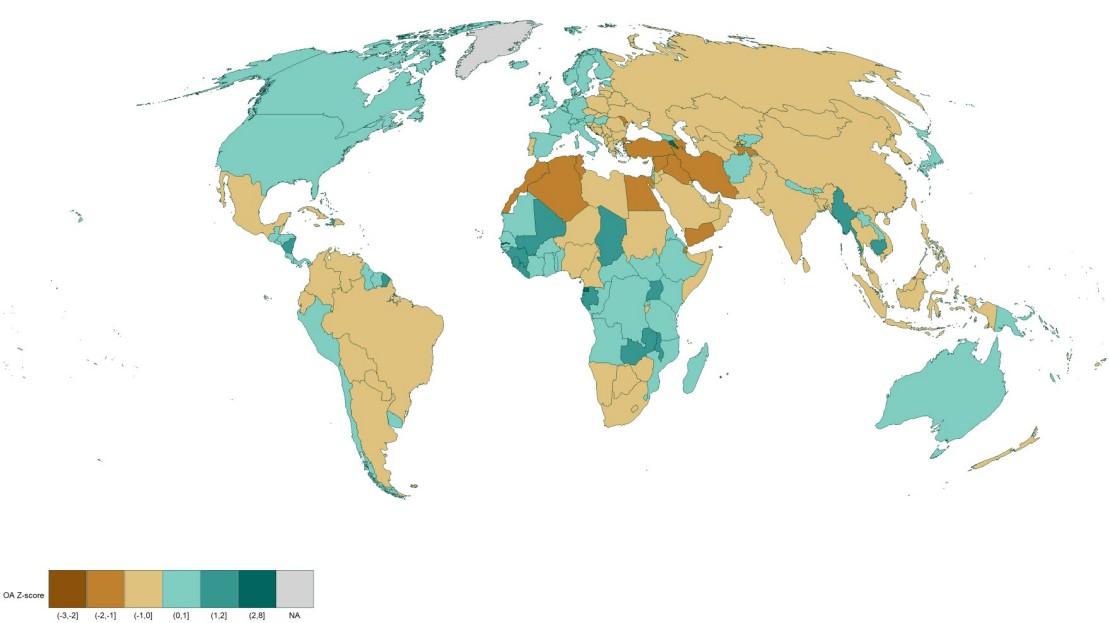

OA Z-score
(-3,-2] (-2,-1] (-1,0] (0,1] (1,2] (2,8] NA

**Fig 2. Weighted z-score and normalized map of references made to OA publications by country.** Red indicates that a country above the world average, blue indicates it is below the world average. White represents the world average. Contains information from OpenStreetMap and OpenStreetMap Foundation, which is made available under the Open Database License.

quadrant. On the other hand, low-income and lower-middle-income countries are generally located in the first quadrant. This shows that overall, lower-income countries tend to both publish and cite OA more than the rest of the world, while the upper-middle-income countries are generally publishing and citing OA under the world average. Open access practices tend to vary a lot from a country to another in high-income countries, with about half being over the average and the other half being under.

Fig 5 shows the average weighted z-score for green and gold OA publications and references to green and gold OA publications by income categories. It shows that while low-income countries tend to publish and cite both green and gold OA, high-income countries mostly use green OA while considerably underusing gold OA. Looking at middle-income countries, lower-middle-income countries tend to use gold OA more than green OA, while upper-middle-income countries underuse both types of OA. S1 Appendix shows detailed world maps of gold and green OA publications and reference patterns.

## Discussion

Our findings show that for the period between 2015 and 2019, Sub-Saharan Africa is publishing and citing OA at a higher rate than the rest of the world, while we found that the Middle East and Asia are the areas where the proportion of publications available in OA is lower, as is their use of OA. This could be explained by the fact that APCs are often waived for low-income countries, such as most Sub-Saharan countries (e.g. through the Research4Life program), but are not or only partially waived for middle-income countries such as the ones found in the Middle East and Asia. One other possible explanation is that the national and transnational OA initiatives such as Plan S for the European Union and PubMed Central in the United States are almost non-existent in these countries, and the Registry of Open Access Repositories shows that institutional repositories are not well developed. Looking at income levels, our

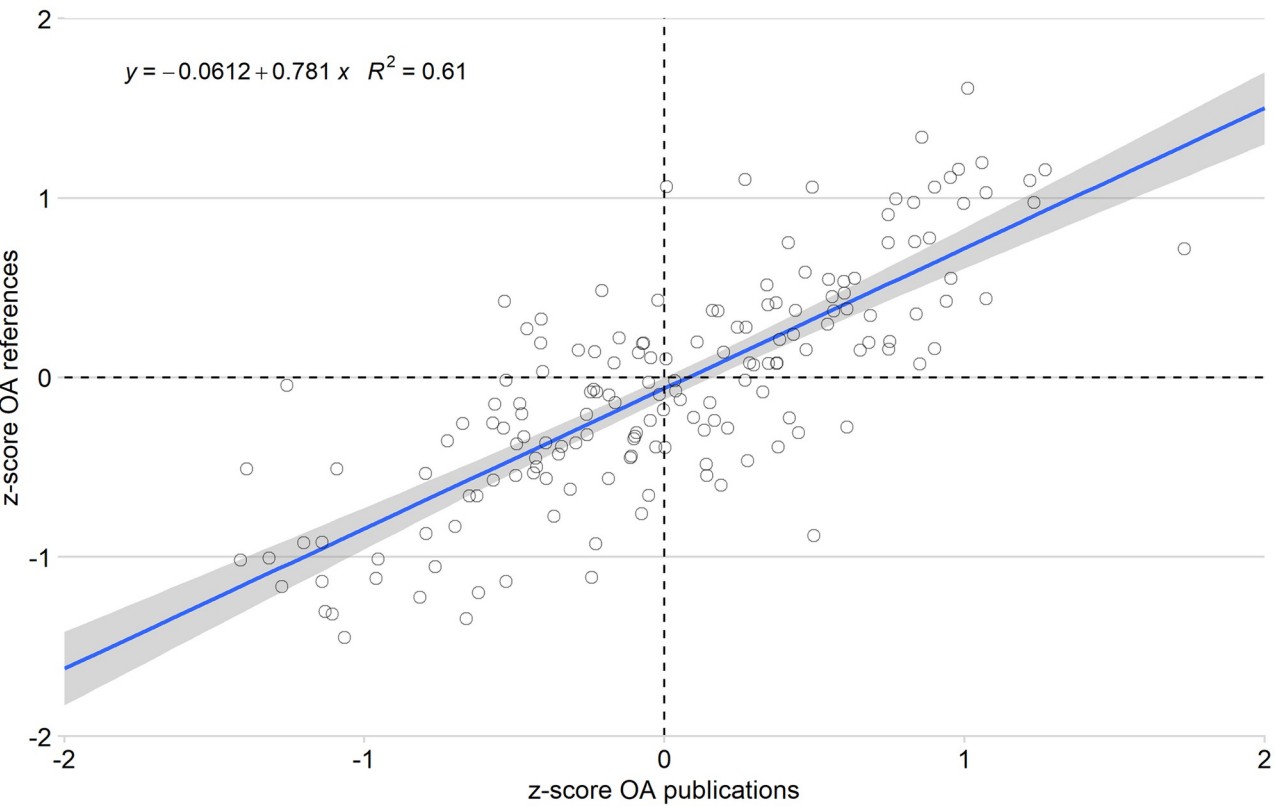

**Fig 3. Correlation between normalized OA publications and normalized cited references by country.**

findings also reveal that lower middle-income and low-income countries are those who publish and cite the most OA. Again, this may be partly explained by the fact that APCs are generally waived for these countries, making publishing in gold OA more accessible for them. Moreover, some commercially owned gold OA journals may have less strict publication criteria and given the lack of research infrastructure in these countries, they may be more likely to publish in these journals [32]. We also found that the upper-middle-income countries tend to behave similarly to the higher-income countries. However, the underlying mechanisms behind the use of OA potential may be different. For instance, upper-middle-income countries may lack the resources to pay for APCs on top of high subscription prices for closed journals (both don't qualify for waivers), which is not necessarily the case for high-income countries. There may also be other factors, such as a lower reputation of OA journals in certain parts of the world. Another interesting finding is that while low-income countries use both green and gold OA, high-income countries generally tend to favour green OA. This may imply that even in the wealthiest countries on the planet, researchers may still struggle to pay APCs or be opposed to them. A possible limitation of the study is our use of the DOIs to identify the papers, given that the number of journals using DOIs might not be evenly distributed around the world, which could have an effect on our analyses of certain countries. Another potential limitation is the use of the Web of Science as the main data source. A study by Basson et al. [33] underlined the importance of data sources and their effects on the measurement of OA, especially when assessing the global South, where a large body of the published research may not be indexed in the most popular bibliometric data sources such as Scopus or Web of Science. Overall, our

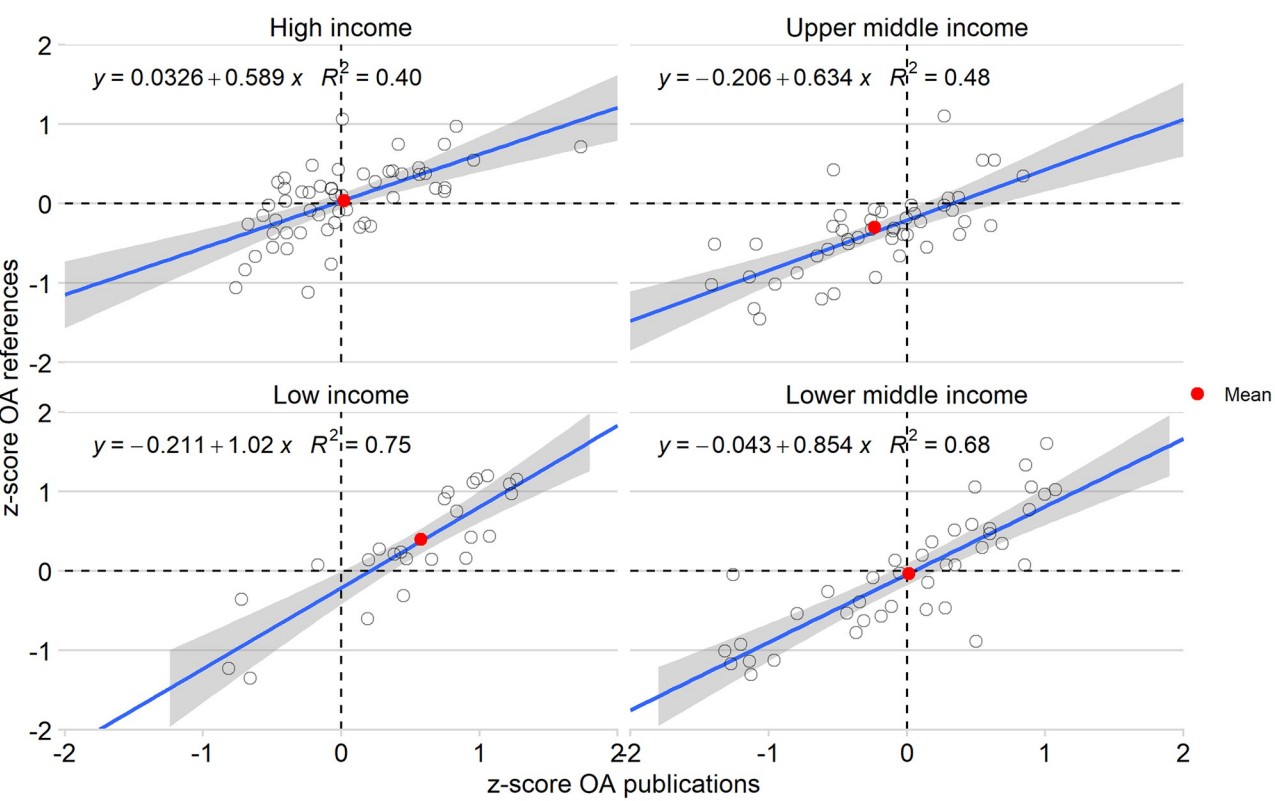

**Fig 4. OA publication vs. references per country income (red dots; average for all the countries of the same income category).**

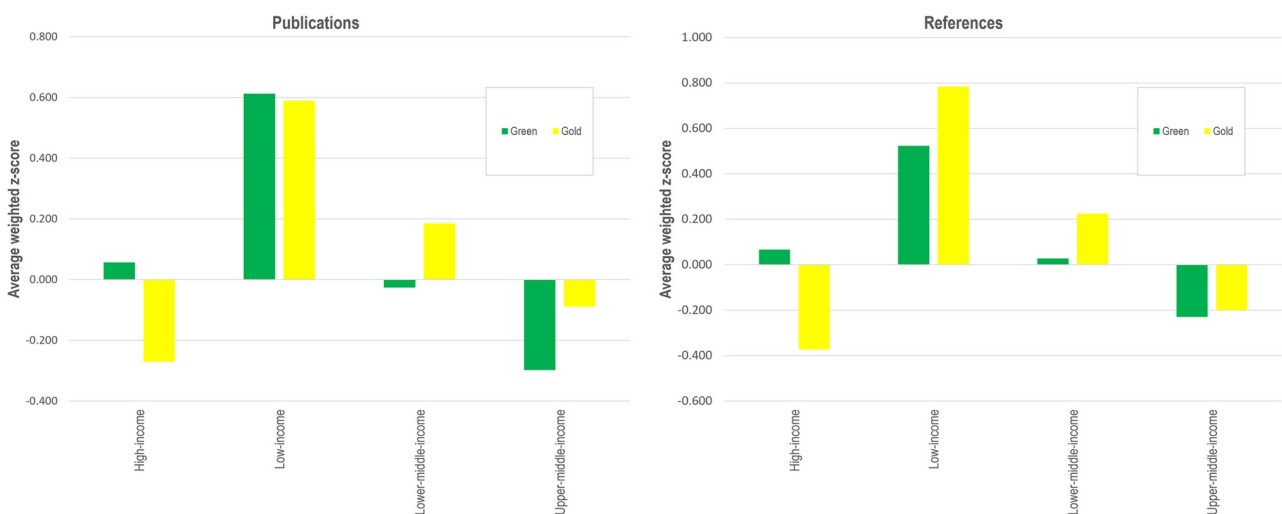

**Fig 5. Average weighted z-score for green and gold OA publications (left) and references to green and gold OA publications (right) by income category.**

results seem to confirm previous studies [9, 10], which found that low-income countries tend to publish in OA more than other countries. Ultimately, while high-income countries have mandates and repositories, and low-income countries have waivers, our results highlight national differences in OA uptake and suggest that more OA initiatives at the institutional, national, and international levels are needed to support wider adoption of open scholarship.

## Conclusion

Our study has provided a snapshot of the state of OA over the 2015–2019 period. While a significant share of the scientific output was available in open access at the time of this study, concerns have been raised about the preservation of open access literature and the integrity of the scholarly communication record [34]. Combining several bibliographic indexes such as Scopus, Ulrichsweb, and the Directory of Open Access Journals, the authors traced journals through the Internet Archive's Wayback Machine. Their results showed that 174 OA journals from various disciplines and geographic regions had completely disappeared from the web between 2000 and 2019. While this number does not appear to be particularly high, the authors noted that their results were most likely a low estimation since the data availability limitations did not allow them to assess the full extent of the phenomenon. They also noticed some common traits between the vanished journals: they tended to be affiliated with academic institutions or scholarly societies located in North America or published social sciences and humanities research. Ultimately, their study underlined the importance of collaborative action to ensure continued access and prevent further scholarly knowledge loss. It remains uncertain how this may play a part in the general uptake and sustainability of OA. Further research on this topic is certainly needed.

The recent COVID-19 pandemic has underlined the importance of OA, which has proven essential for faster and more efficient dissemination and use of scholarly literature. At the journal level, several publishers such as Springer-Nature, AAAS (Science), Elsevier, and Massachusetts Medical Society (NEJM) have announced the free opening of their COVID-19-related papers. However, it remains unclear if these papers will remain OA. Elsevier, for instance, already mentioned that their Novel Coronavirus Information Center will only be free for the duration of the pandemic [35]. Despite the existence of geopolitical tensions [36–38] and initiatives such as China's move to ensure that funded studies on COVID-19 will be published in Chinese journals over international journals [39], the majority of countries have increased their number of international collaborations and have generally increased their share of OA literature [38, 40]. However, it has been shown that countries' previous OA dissemination practices had no significant effect on their OA publication practices during the ongoing global pandemic: the extent to which a country was affected by COVID-19 was the main factor for OA publication [38]. Another important finding from that study is that countries with a higher GDP may not engage as much as other countries in international collaboration and open access during the global crisis, most likely because they are less dependent on outside collaborators. These findings have several implications for our study. First, the pandemic may help middle-income countries to participate and alleviate the divide in the uptake of OA since they do not necessarily have the necessary funding and resources to produce national COVID-19 research. Second, with high-income countries being more reluctant to participate in global collaboration and publish their COVID-19 research in OA, it highlights the importance of the various OA mandates, policies, and repositories in place. It remains to be seen whether this recent trend created by the pandemic will persist over time. Ultimately, the current crisis shows the necessity for more sustainable OA dissemination channels: as long as the research community is dependent on private for-profit publishers, the "openness" of articles will

depend on either the "good will" of those publishers or on the scientific community's capacity and willingness to pay for it.

## Supporting information

**S1 Appendix. Weighted z-score and normalized map of the number of OA publications and references made to OA publications by country.** Red indicates that a country is above the world average, blue indicates it is below the world average. White represents the world average. Contains information from OpenStreetMap and OpenStreetMap Foundation, which is made available under the Open Database License.
(TIF)

## Acknowledgments

The authors would like to thank Vanessa Sandoval-Romero for her participation in the initial phases of the project. We would also like the thank Marion Maisonobe and Iryna Kuchma for their time and their invaluable comments and suggestions. This article is a revised version of an ISSI paper with updated data that was originally published in the Proceedings of the 18th International Conference on Scientometrics and Informetrics.

## Author Contributions

**Conceptualization:** Marc-André Simard, Gita Ghiasi, Philippe Mongeon, Vincent Larivière.

**Data curation:** Philippe Mongeon.

**Formal analysis:** Marc-André Simard, Gita Ghiasi, Philippe Mongeon.

**Funding acquisition:** Marc-André Simard, Vincent Larivière.

**Investigation:** Marc-André Simard, Gita Ghiasi, Philippe Mongeon.

**Methodology:** Marc-André Simard, Gita Ghiasi, Philippe Mongeon.

**Project administration:** Marc-André Simard.

**Resources:** Vincent Larivière.

**Software:** Marc-André Simard, Gita Ghiasi, Philippe Mongeon.

**Supervision:** Vincent Larivière.

**Validation:** Philippe Mongeon.

**Visualization:** Marc-André Simard, Philippe Mongeon.

**Writing – original draft:** Marc-André Simard, Gita Ghiasi, Philippe Mongeon.

**Writing – review & editing:** Marc-André Simard, Gita Ghiasi, Philippe Mongeon, Vincent Larivière.

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
