## [Decision Letter · Decision Letter 0]

3 Nov 2021

PONE-D-21-29770National differences in dissemination and use of open access literaturePLOS ONE

Dear Dr. Simard,

Thank you for submitting your manuscript to PLOS ONE. After careful consideration, we feel that it has merit but does not fully meet PLOS ONE’s publication criteria as it currently stands. Therefore, we invite you to submit a revised version of the manuscript that addresses the points raised during the review process.

I agree with the two reviewers about the importance of your descriptive analysis. But I think that the paper should be revised before publication.

The first and most problematic point is about data availability, as agreed by two reviewers. The main contribution of the paper is connected to the descriptive analysis of a very large database.  But the underlying data, classifications and indicators adopted are not appropriately described and released. The paper appears not to be compliant with data availability policy as stated here: https://journals.plos.org/plosone/s/data-availability.

I think that there are two possible solutions to overcome this problem. The first one is to release anonymized raw data containing only the minimum set of information useful for replicating the analysis. For example, if you limit the set of metadata to an anonymous ID, country, type of open access, discipline/fields I think that this is not in conflict with the legal restrictions imposed by Clarivate. As an alternative or better as a complement to the raw data you should release the aggregated and reworked data.

I agree with Reviewer 2 that the presentation of data is not completely satisfactory, and I would suggest a better refinement of table and figures. Moreover, I think that the choice of omitting all computational details is a major shortcoming of the paper. For example, no clear definitions of fields/discipline classification of papers and normalizations are available for readers. Analogously readers are unaware of the classification of countries. I think that readers would benefit by a better documentation of your analysis.

We look forward to receiving your revised manuscript.

Kind regards,

Alberto Baccini, Ph.D.

Academic Editor

PLOS ONE

Journal Requirements:

“YES - Vincent Larivière would like you acknowledge the Canada Research Chair program (grant 950-231768). Marc-André Simard would like to acknowledge the SSHRC Joseph Armand Bombardier Master’s Scholarship.”

“No”

5. We note that Figures 1a, 2a, appendix1a and appendix1b in your submission contain [map/satellite] images which may be copyrighted. All PLOS content is published under the Creative Commons Attribution License (CC BY 4.0), which means that the manuscript, images, and Supporting Information files will be freely available online, and any third party is permitted to access, download, copy, distribute, and use these materials in any way, even commercially, with proper attribution. For these reasons, we cannot publish previously copyrighted maps or satellite images created using proprietary data, such as Google software (Google Maps, Street View, and Earth). For more information, see our copyright guidelines: http://journals.plos.org/plosone/s/licenses-and-copyright.

a. You may seek permission from the original copyright holder of Figures 1a, 2a, appendix1a and appendix1b to publish the content specifically under the CC BY 4.0 license. 

6. We noted in your submission details that a portion of your manuscript may have been presented or published elsewhere. Please clarify whether this publication was peer-reviewed and formally published. If this work was previously peer-reviewed and published, in the cover letter please provide the reason that this work does not constitute dual publication and should be included in the current manuscript.

Reviewers' comments:

Reviewer's Responses to Questions

**Comments to the Author**

1. Is the manuscript technically sound, and do the data support the conclusions?

Reviewer #1: Yes

Reviewer #2: Partly

2. Has the statistical analysis been performed appropriately and rigorously? 

Reviewer #1: Yes

Reviewer #2: Yes

3. Have the authors made all data underlying the findings in their manuscript fully available?

Reviewer #1: No

Reviewer #2: No

4. Is the manuscript presented in an intelligible fashion and written in standard English?

Reviewer #1: Yes

Reviewer #2: Yes

5. Review Comments to the Author

Reviewer #1: Thank you for this timely and very interesting paper! Please see some comments and suggestions below.

I would add at low or no cost to this sentence in the Abstract: as well as the development of several new platforms that facilitate the publication of OA content at a low cost.

Background - Open access models - I am not sure ROAR data is up to date. I would use OpenDOAR for current repository statistics https://v2.sherpa.ac.uk/opendoar/ - 5743 repositories as of October 2021 https://v2.sherpa.ac.uk/view/repository_visualisations/1.html. And I would revise this sentence with data from OpenDOAR: "There are currently 4,607 repositories listed in the Registry of Open Access Repositories (ROAR), including 1,021 in North America (839 in the United States,), 1702 in Europe, 179 in Africa, 1021 in Asia, and 584 in South America."

I am not sure whether rumours is the accurate word in the sentence below, perhaps it could be replaced with discussions or something similar - In the United States, there have been rumors of a new OA initiative comparable to Plan S in Europe: this policy would make every...

Perhaps it's also worth mentioning Redalyc https://www.redalyc.org/ and AmeliCA http://amelica.org/ together with SciELO publishing platforms? And mention that SciELO collaborates with South Africa http://www.scielo.org.za/? And what about repositories platforms - e.g. LA Referencia in Latin America https://www.lareferencia.info/en/, CAS IR Grid in China http://www.irgrid.ac.cn/?

And I am not sure how accurate is this sentence in the Conclusion: This could be explained by the fact that APCs are mostly waived for Sub-Saharan countries (e.g. research4life countries). Perhaps you could add that APCs are mostly waived for low income countries and discounted for low-middle income countries? E.g. from Taylor and Francis https://authorservices.taylorandfrancis.com/publishing-open-access/oa-funding-options/.

I would be curious to see how many of OA journals in your study didn't charge APCs - APCs waivers have been mentioned as one of the reasons for OA publishing, but perhaps these were the journals that didn't charge APCs.

And perhaps the UNESCO reference below could be updated with the latest version of the Open Science Recommendation that governments will vote on in November: https://unesdoc.unesco.org/ark:/48223/pf0000378381.locale=en

27. UNESCO. Towards a UNESCO Recommendation on Open Science [Internet]. 2020

[cited 2021 Jul 16]. Available from:

https://en.unesco.org/sites/default/files/open_science_brochure_en.pdf

Reviewer #2: The manuscript is very clear and well presented and the data analysis has been done seriously. The corpus used is also clearly described and the literature review is quite well provided and informative. For all these qualities, it seems to me that the article deserves only minor revisions. That said, the article does have weaknesses and would benefit from substantial improvements. I provide in the following a set of comments made during the reading, which you will find in the uploaded version of the manuscript with comments. But before that, I think it is important to stress the two main weaknesses of the article:

- First, the result section is very brief and limited to simple comments of graphs and maps. All the non data-related material is in the long introduction and background and in the conclusion. This is perhaps a choice of presentation and I would be ready to accept it if it is justified, but from a rhetorical point of view, it seems to me that the contribution of the article would perhaps be more convincing with an enriched result section, why not with the help of some arguments and interpretations that are actually stated elsewhere in the article.

- Secondly, the maps and graphs suffer from many flaws and need to be reworked. This is all the more inconvenient as they occupy a central place in the result section and are therefore essential elements in the purpose and contribution of this article.

Finally, the raw data cannot be shared because of the agreement with Clarivates, but the aggregated and reworked data (Z-score at country level) are certainly not affected by this same restriction and would benefit from being openly shared.

In what follows, I report the comments made in a linear way when reading the article and which you can also find in the commented version of the manuscript:

Page 5: As you processed the data and have indicators at the global scale, I believe you could share the Z-scores you computed at the country level

Page 10 :

- "with the creation of a dozen more journals"  "with the creation of a dozen more electronic journals"

- You should introduce the plan of the article at the end of the introduction (before the backgroud section)

Page 13 :missing word before "at": "In terms of OA usage by different at the country level level"

Page 14, last paragraph:Note that the result of the paper by Iyandemye and Thomas is already cited in a previous paragraph.

Page 16, first paragraph: Note that the NIH policy is already mentioned earlier in the paper.

Page 16, first paragraph: "has contributed to the expensive of PMC"  "to the expansion of PMC"

Page 18, first paragraph: Given that the number of journals with DOIs may not be very evenly distributed around the world, I think that ignoring articles without DOIs could skew your analysis. If you think so too, I think it would be worth mentioning this limitation.

Page 18, first paragraph: "This data indicates whether the paper is published in OA or not, and in the OA category it belongs to"  "and in the OA category it belongs to" (useless "in")

Page 18, second paragraph, first sentence: useless "that"

Page 18, last paragraph: less OA papers in the Humanities than in Medecine  This result must be interpreted with caution since the type of social sciences and humanities journals indexed on the Web of Science are mostly owned by for profit publishers such as Wiley. In other words, dont you think the WoS coverage bias can affect this observation?

Page 20, first sentence: "Figures 1 and 2 reveal that countries mostly cite OA more often than they publish in OA."  You latter show that this really depends on the countries. Therefore this first sentence should be rewritten. If you really want to state that "most countries cite OA more often than they publish in OA", then I advise you to use a global statistics i.e. the % of countries citing OA more often than they publish in OA.

Page 20, last sentence: It would be clearer for the reader if you specify in parenthesis after both indicator what is exactly the variable that you used (I believe it is thez-score mentioned in the method section but helping the reader would be useful here).

Page 21, first sentence: The relation is not that strong at the global level indeed, justifying to look at the divergence between groups of countries regarding this statistical relation as you have done below.

Page 22: "This could be explained by the fact that APCs are mostly waived for Sub-Saharan countries (e.g. research4life countries) but are not or only partially waived for most Middle Eastern and Asian countries"  Therefore, It could have been interesting to consider groups of countries according to these differences instead of using the world bank's classification

Page 23: "This may imply that even in the richest countries on the planet, researchers may still struggle to pay APCs."  Or be opposed to them?

Page 24: "Despite the existence of geological tensions"  "geopolitical" instead of "geological"

Page 31: Figure 1: This map is blur and the projection must be changed (the mercator projection is only relevant when considering sea travels since it preserves navigation angles) : https://www.britannica.com/science/Mercator-projection

Page 32: The legend should be on the map and a precision on the unit of measure should be added

Page 33 and 34: Figure 2: same remarks as for Figure 1a and Figure 1b

Page 35, Figure 3: This graphic could be much more interesting and self-explanatory in labelling the nodes and in adding the formula of the regression line in a box. See for instance with R: https://www.r-graph-gallery.com/web-scatterplot-corruption-and-human-development.html and: https://rpkgs.datanovia.com/ggpubr/reference/stat_regline_equation.html

Page 36, Figure 4: Same remark as for the previous plot, the quality of these plots is very low. In addition, the dots are too bigs which creates a lot of superpositions.

Page 37, Figure 5: Blur but fine

Page 38, appendix: The projection is incorrect and the legend is not explanatory (the unit of measure is missing)

6. PLOS authors have the option to publish the peer review history of their article (what does this mean?). If published, this will include your full peer review and any attached files.

Reviewer #1: **Yes: **Iryna Kuchma

Reviewer #2: **Yes: **Marion Maisonobe

---

## [Author Response · Author response to Decision Letter 0]

24 May 2022

Editor comments:

The first and most problematic point is about data availability, as agreed by two reviewers. The main contribution of the paper is connected to the descriptive analysis of a very large database. But the underlying data, classifications and indicators adopted are not appropriately described and released. The paper appears not to be compliant with data availability policy as stated here: https://journals.plos.org/plosone/s/data-availability.

I think that there are two possible solutions to overcome this problem. The first one is to release anonymized raw data containing only the minimum set of information useful for replicating the analysis. For example, if you limit the set of metadata to an anonymous ID, country, type of open access, discipline/fields I think that this is not in conflict with the legal restrictions imposed by Clarivate. As an alternative or better as a complement to the raw data you should release the aggregated and reworked data.

I agree with Reviewer 2 that the presentation of data is not completely satisfactory, and I would suggest a better refinement of table and figures. Moreover, I think that the choice of omitting all computational details is a major shortcoming of the paper. For example, no clear definitions of fields/discipline classification of papers and normalizations are available for readers. Analogously readers are unaware of the classification of countries. I think that readers would benefit by a better documentation of your analysis.

Answer: We have added some information about the computational details and updated all of the figures. The data used is now also available. 

Journal Requirements:

“YES - Vincent Larivière would like to acknowledge the Canada Research Chair program (grant 950-231768). Marc-André Simard would like to acknowledge the SSHRC Joseph Armand Bombardier Master’s Scholarship.”

“No”

Answer: We have shared a dataset that allows reproduction of the analyses. 

5. We note that Figures 1a, 2a, appendix1a and appendix1b in your submission contain [map/satellite] images which may be copyrighted. All PLOS content is published under the Creative Commons Attribution License (CC BY 4.0), which means that the manuscript, images, and Supporting Information files will be freely available online, and any third party is permitted to access, download, copy, distribute, and use these materials in any way, even commercially, with proper attribution. For these reasons, we cannot publish previously copyrighted maps or satellite images created using proprietary data, such as Google software (Google Maps, Street View, and Earth). For more information, see our copyright guidelines: http://journals.plos.org/plosone/s/licenses-and-copyright.

a. You may seek permission from the original copyright holder of Figures 1a, 2a, appendix1a and appendix1b to publish the content specifically under the CC BY 4.0 license. 

Answer: The software we used for the maps (Tableau) uses maps from Open Street Map (see copyright notice in the bottom of our map) which has an open license: 

https://www.openstreetmap.org/

6. We noted in your submission details that a portion of your manuscript may have been presented or published elsewhere. Please clarify whether this publication was peer-reviewed and formally published. If this work was previously peer-reviewed and published, in the cover letter please provide the reason that this work does not constitute dual publication and should be included in the current manuscript.

Answer: The original version was peer-reviewed and published as a part of the Proceedings of the 2021 International Conference on Scientometrics and Informetrics (ISSI, 2021) as a work in progress. The original version used older data and had less than 5 pages of content (can be found here: https://tinyurl.com/MASOA). For the new version, we updated the data and added more to every individual section. 

Comments 

Reviewer #1: 

Thank you for this timely and very interesting paper! Please see some comments and suggestions below.

I would add at low or no cost to this sentence in the Abstract: as well as the development of several new platforms that facilitate the publication of OA content at a low cost.

Answer: Thank you for the suggestion, we modified the sentence as suggested.

Background - Open access models - I am not sure ROAR data is up to date. I would use OpenDOAR for current repository statistics https://v2.sherpa.ac.uk/opendoar/ - 5743 repositories as of October 2021 https://v2.sherpa.ac.uk/view/repository_visualisations/1.html. And I would revise this sentence with data from OpenDOAR: "There are currently 4,607 repositories listed in the Registry of Open Access Repositories (ROAR), including 1,021 in North America (839 in the United States,), 1702 in Europe, 179 in Africa, 1021 in Asia, and 584 in South America."

Answer: Thank you, we added the number of repositories reported in OpenDOAR alongside the updated numbers from the ROAR. We kept the ROAR data since the registry provides the numbers by continent, which we find useful in the context of this study. The text now reads as follows:

As of November 6 2021, there were 5,764 and 4,629 repositories listed in the Directory of Open Access Repositories (OpenDOAR) and the Registry of Open Access Repositories (ROAR), respectively. The repositories are distributed as follows according to the ROAR data: Europe (36.8%), Asia (22.3%), North America (22.1%), South America (12.8%), Africa (3.9%), and Oceania (2.1%). 

I am not sure whether rumours is the accurate word in the sentence below, perhaps it could be replaced with discussions or something similar - In the United States, there have been rumors of a new OA initiative comparable to Plan S in Europe: this policy would make every…

Answer: Thank you, we modified the sentence and used “discussions” instead.

Perhaps it's also worth mentioning Redalyc https://www.redalyc.org/ and AmeliCA http://amelica.org/ together with SciELO publishing platforms? And mention that SciELO collaborates with South Africa http://www.scielo.org.za/? And what about repositories platforms - e.g. LA Referencia in Latin America https://www.lareferencia.info/en/, CAS IR Grid in China http://www.irgrid.ac.cn/?

Answer: Thank you. All of the suggestions were added.

And I am not sure how accurate is this sentence in the Conclusion: This could be explained by the fact that APCs are mostly waived for Sub-Saharan countries (e.g. research4life countries). Perhaps you could add that APCs are mostly waived for low income countries and discounted for low-middle income countries? E.g. from Taylor and Francis https://authorservices.taylorandfrancis.com/publishing-open-access/oa-funding-options/.

Answer: We have added the income levels in order to make the statement more accurate. Thank you.

I would be curious to see how many of OA journals in your study didn't charge APCs - APCs waivers have been mentioned as one of the reasons for OA publishing, but perhaps these were the journals that didn't charge APCs.

And perhaps the UNESCO reference below could be updated with the latest version of the Open Science Recommendation that governments will vote on in November: https://unesdoc.unesco.org/ark:/48223/pf0000378381.locale=en

27. UNESCO. Towards a UNESCO Recommendation on Open Science [Internet]. 2020

[cited 2021 Jul 16]. Available from:

https://en.unesco.org/sites/default/files/open_science_brochure_en.pdf

Answer: Thank you. We have updated the reference and added a sentence on the official adoption of the Recommendation. 

Reviewer #2: 

The manuscript is very clear and well presented and the data analysis has been done seriously. The corpus used is also clearly described and the literature review is quite well provided and informative. For all these qualities, it seems to me that the article deserves only minor revisions. That said, the article does have weaknesses and would benefit from substantial improvements. I provide in the following a set of comments made during the reading, which you will find in the uploaded version of the manuscript with comments. But before that, I think it is important to stress the two main weaknesses of the article:

- First, the result section is very brief and limited to simple comments of graphs and maps. All the non data-related material is in the long introduction and background and in the conclusion. This is perhaps a choice of presentation and I would be ready to accept it if it is justified, but from a rhetorical point of view, it seems to me that the contribution of the article would perhaps be more convincing with an enriched result section, why not with the help of some arguments and interpretations that are actually stated elsewhere in the article.

Answer: Thank you for the comment. Indeed, we did not comment on the results within the results section on purpose. We chose to describe (results) and then discuss in the now renamed discussion and conclusion section.

- Secondly, the maps and graphs suffer from many flaws and need to be reworked. This is all the more inconvenient as they occupy a central place in the result section and are therefore essential elements in the purpose and contribution of this article.

Answer: The file conversion process had a very negative effect on the resolution of the files. When we noticed this, the submission had already been sent. New higher definition versions of the figures have been added. Thank you. 

Finally, the raw data cannot be shared because of the agreement with Clarivates, but the aggregated and reworked data (Z-score at country level) are certainly not affected by this same restriction and would benefit from being openly shared.

Answer: We prepared a dataset that can be shared. Thank you. 

In what follows, I report the comments made in a linear way when reading the article and which you can also find in the commented version of the manuscript:

Page 5: As you processed the data and have indicators at the global scale, I believe you could share the Z-scores you computed at the country level

Answer: The Z-scores will be a part of the new dataset. Thank you. 

Page 10 :

- "with the creation of a dozen more journals"  "with the creation of a dozen more electronic journals"

Answer: Added, thank you. 

- You should introduce the plan of the article at the end of the introduction (before the background section)

Page 13 :missing word before "at": "In terms of OA usage by different at the country level"

Answer: Thank you, we fixed the sentence.

Page 14, last paragraph:Note that the result of the paper by Iyandemye and Thomas is already cited in a previous paragraph.

Answer: Thank you, we modified the sentence to only refer to the part of the Iyandemye and Thomas results that relate to this paragraph, and thus removed the repetition.

Page 16, first paragraph: Note that the NIH policy is already mentioned earlier in the paper.

Answer: Thank you, we removed the second mention of the NIH policy.

Page 16, first paragraph: "has contributed to the expensive of PMC"  "to the expansion of PMC"

Answer: Thank you, we made the correction.

Page 18, first paragraph: Given that the number of journals with DOIs may not be very evenly distributed around the world, I think that ignoring articles without DOIs could skew your analysis. If you think so too, I think it would be worth mentioning this limitation.

Answer: Thank you. We have added a sentence to mention this limitation. 

Page 18, first paragraph: "This data indicates whether the paper is published in OA or not, and in the OA category it belongs to"  "and in the OA category it belongs to" (useless "in") 

Answer: Thank you, we made the correction.

Page 18, second paragraph, first sentence: useless "that"

Answer: Thank you, we made the correction.

Page 18, last paragraph: less OA papers in the Humanities than in Medecine  This result must be interpreted with caution since the type of social sciences and humanities journals indexed on the Web of Science are mostly owned by for profit publishers such as Wiley. In other words, don't you think the WoS coverage bias can affect this observation?

Answer: The data source is definitely one of the limits of this study. We have added a sentence to reflect this situation in the discussion. Thank you. 

Page 20, first sentence: "Figures 1 and 2 reveal that countries mostly cite OA more often than they publish in OA."  You latter show that this really depends on the countries. Therefore this first sentence should be rewritten. If you really want to state that "most countries cite OA more often than they publish in OA", then I advise you to use a global statistics i.e. the % of countries citing OA more often than they publish in OA.

Answer: We removed this sentence as it was indeed problematic. Thank you. 

Page 20, last sentence: It would be clearer for the reader if you specify in parenthesis after both indicator what is exactly the variable that you used (I believe it is the z-score mentioned in the method section but helping the reader would be useful here).

Answer: We have added the % of references and OA publications that the text referred to. Thank you. 

Page 21, first sentence: The relation is not that strong at the global level indeed, justifying to look at the divergence between groups of countries regarding this statistical relation as you have done below.

Page 22: "This could be explained by the fact that APCs are mostly waived for Sub-Saharan countries (e.g. research4life countries) but are not or only partially waived for most Middle Eastern and Asian countries"  Therefore, It could have been interesting to consider groups of countries according to these differences instead of using the world bank's classification

Answer: That is a good point. However, we felt that using the World Bank’s classification gave us a good “frame” for this. We also wanted our results to be comparable to those of Evans and Reimer (2009) who also used the World Bank’s classification for countries. 

Page 23: "This may imply that even in the richest countries on the planet, researchers may still struggle to pay APCs."  Or be opposed to them?

Answer: Or both! Thank you, we added this to the paper.

Page 24: "Despite the existence of geological tensions"  "geopolitical" instead of "geological"

Answer: Thank you, we made the correction.

Page 31: Figure 1: This map is blur and the projection must be changed (the mercator projection is only relevant when considering sea travels since it preserves navigation angles) : https://www.britannica.com/science/Mercator-projection

Answer: The software we used to generate the maps (Tableau) does not seem to allow different types of projection. 

Page 32: The legend should be on the map and a precision on the unit of measure should be added

Answer: We fixed the map and the legend, thank you. 

Page 33 and 34: Figure 2: same remarks as for Figure 1a and Figure 1b

Answer: We fixed the map and the legend, thank you

Page 35, Figure 3: This graphic could be much more interesting and self-explanatory in labelling the nodes and in adding the formula of the regression line in a box. See for instance with R: https://www.r-graph-gallery.com/web-scatterplot-corruption-and-human-development.html and: https://rpkgs.datanovia.com/ggpubr/reference/stat_regline_equation.html

Answer: Thank you, we redid the figures in R to make them nicer and clearer.

Page 36, Figure 4: Same remark as for the previous plot, the quality of these plots is very low. In addition, the dots are too big which creates a lot of superpositions.

Answer: Thank you, we redid the figures in R to make them nicer and clearer.

Page 37, Figure 5: Blur but fine

Answer: We increased the resolution of the figure to resolve this. Thank you. 

Page 38, appendix: The projection is incorrect and the legend is not explanatory (the unit of measure is missing)

Answer: We fixed the maps and the legends, thank you. However, as previously stated, the software we used (Tableau) does not seem to allow the use of different projections.

---

## [Decision Letter · Decision Letter 1]

27 Jun 2022

PONE-D-21-29770R1National differences in dissemination and use of open access literaturePLOS ONE

Dear Dr. Simard,

Thank you for submitting your manuscript to PLOS ONE. After careful consideration, we feel that it has merit but does not fully meet PLOS ONE’s publication criteria as it currently stands. Therefore, we invite you to submit a revised version of the manuscript that addresses the points raised during the review process. In my opinion, the suggestions of Reviewer 2 may be useful for improving the quality of your paper. I think however that it is in your complete responsability as authors to accept them or not. 

We look forward to receiving your revised manuscript.

Kind regards,

Alberto Baccini, Ph.D.

Academic Editor

PLOS ONE

Journal Requirements:

Reviewers' comments:

Reviewer's Responses to Questions

**Comments to the Author**

1. If the authors have adequately addressed your comments raised in a previous round of review and you feel that this manuscript is now acceptable for publication, you may indicate that here to bypass the “Comments to the Author” section, enter your conflict of interest statement in the “Confidential to Editor” section, and submit your "Accept" recommendation.

Reviewer #1: All comments have been addressed

Reviewer #2: (No Response)

2. Is the manuscript technically sound, and do the data support the conclusions?

Reviewer #1: Yes

Reviewer #2: Yes

3. Has the statistical analysis been performed appropriately and rigorously? 

Reviewer #1: Yes

Reviewer #2: Yes

4. Have the authors made all data underlying the findings in their manuscript fully available?

Reviewer #1: Yes

Reviewer #2: Yes

5. Is the manuscript presented in an intelligible fashion and written in standard English?

Reviewer #1: Yes

Reviewer #2: Yes

6. Review Comments to the Author

Reviewer #1: (No Response)

Reviewer #2: I would like to thank the authors for the new version of their article and especially for the open access and sharing of the data. This gives a real added value to their contribution.

Despite the changes made, I have identified a number of shortcomings in the revised text that will need to be corrected before the article can be published.

1. Contrary to what the authors have indicated, the repetition regarding the NIH policy is still present in the current version: to avoid this repetition, one option is to move the paragraph on NIH that is in the “OA models” section to the “initiatives and platforms” section (where the NIH policy is introduced).

2. Some references are not properly introduced, which can make them seem irrelevant or confusing (they are listed below).

3. Contrary to what was requested, the maps have not been redone. To solve this issue, we provide the authors with a script and a figure adapted from their data that they can reuse to generate Figure 1 and 2. We show in this script that it is possible to make a discretization of the z-score variable rather than representing it as a continuous variable. This approach allows to obtain a clearer figure and to better manage extreme values such as the score of Equatorial Guinea (in the authors' version, it seems that it has been removed (as the max is 2 on the legend) which is problematic). We also note that the data is missing for Greenland (given that there is a university in Greenland, is this due to missing data or an omission?)

4. Finally, we have a suggestion to make to improve the readability of the 1st part. As the first part of the article lists a large number of facts that made it possible to reconstruct a chronology of the development of Open Access: on re-reading, we think that it would be helpful to add a timeline or a summary listing the main events mentioned in the text in chronological order. This will facilitate the reading of the text. If the authors do not wish to add a new figure in the body of the text, they can add this timeline in an appendix. As such, it will be an additional result.

In the following we list the changes requested and the problems identified as they appear in the text:

Page 1: Replace "Psycholoquy" by "Psychology"

Page 3: Replace "Bronze OA refers to publications that are openly directly" by "Bronze OA refers to publications that are open directly"

Page 4: Miguel et al. is cited (between Pinowar et al and Basson et al) whereas it seems that contrary to these other references Miguel et al concentrate on the literature dealing with OA which is another matter and should be introduced as such.

Page 4: Replace "their effects of the measurement of open access" by "their effects on the measurement of open access"

Page 5: missing dot before reference number 8 (later in the text, we found that there are sometimes missing spaces before reference numbers: eg. before reference 21)

In the “Initiative and Platforms” section: you should create a new paragraph for each geographic area: “In Europe”, “In Canada”, “In the US”, “In South America”, “In Asia”, “In Africa”. It will improve the clarity of this section.

Page 7, in the paragraph about Canada: The reference to the OJS system (32) seems a bit out of context as the link between this system and OA is not explicated

Page 10: The acronym “BOAI” is mentioned whereas it has never been introduced before

Results section, 1st sentence: Replace "Tables 1 and 2 presents" by "Tables 1 and 2 present"

Table 1 caption: missing parenthesis

Figure 5: on the left part, green is in yellow and gold is in green

Figure 1 and 2 + Appendix: the maps should be redraw with the help of the provided script (it is possible to change the colours if you wish to, and, if necessary, to add an intermediary class around zero to represent countries with z-scores equal to the world average)

7. PLOS authors have the option to publish the peer review history of their article (what does this mean?). If published, this will include your full peer review and any attached files.

Reviewer #1: **Yes: **Iryna Kuchma

Reviewer #2: **Yes: **Marion Maisonobe

---

## [Author Response · Author response to Decision Letter 1]

8 Jul 2022

Reviewer #1: (No Response)

Reviewer #2: I would like to thank the authors for the new version of their article and especially for the open access and sharing of the data. This gives a real added value to their contribution.

Answer: Thank you! 

Despite the changes made, I have identified a number of shortcomings in the revised text that will need to be corrected before the article can be published.

1. Contrary to what the authors have indicated, the repetition regarding the NIH policy is still present in the current version: to avoid this repetition, one option is to move the paragraph on NIH that is in the “OA models” section to the “initiatives and platforms” section (where the NIH policy is introduced).

Answer: Our apologies for the confusion about that part. We have removed it from the paper. Thank you! 

2. Some references are not properly introduced, which can make them seem irrelevant or confusing (they are listed below).

Answer: We have fixed the references based on your suggestions. Thank you! 

3. Contrary to what was requested, the maps have not been redone. To solve this issue, we provide the authors with a script and a figure adapted from their data that they can reuse to generate Figure 1 and 2. We show in this script that it is possible to make a discretization of the z-score variable rather than representing it as a continuous variable. This approach allows to obtain a clearer figure and to better manage extreme values such as the score of Equatorial Guinea (in the authors' version, it seems that it has been removed (as the max is 2 on the legend) which is problematic). We also note that the data is missing for Greenland (given that there is a university in Greenland, is this due to missing data or an omission?)

Answer: We have redone the maps based on your suggestions and code provided. Thank you very much for your kindness. 

4. Finally, we have a suggestion to make to improve the readability of the 1st part. As the first part of the article lists a large number of facts that made it possible to reconstruct a chronology of the development of Open Access: on re-reading, we think that it would be helpful to add a timeline or a summary listing the main events mentioned in the text in chronological order. This will facilitate the reading of the text. If the authors do not wish to add a new figure in the body of the text, they can add this timeline in an appendix. As such, it will be an additional result.

Answer: We thank the reviewer for this valuable suggestion. However, we chose not to include a timeline in the article because of time constraints. 

In the following we list the changes requested and the problems identified as they appear in the text:

Page 1: Replace "Psycholoquy" by "Psychology"

Answer: Psycholoquy is the name of an early OA journal. 

Page 3: Replace "Bronze OA refers to publications that are openly directly" by "Bronze OA refers to publications that are open directly"

Answer: We have changed the sentence. Thank you!

Page 4: Miguel et al. is cited (between Pinowar et al and Basson et al) whereas it seems that contrary to these other references Miguel et al concentrate on the literature dealing with OA which is another matter and should be introduced as such.

Answer: The reference has been moved to the next paragraph and reintroduced in a way that highlights its contribution (countries who contribute to OA as a topic).

Page 4: Replace "their effects of the measurement of open access" by "their effects on the measurement of open access"

Answer: We have changed the sentence. Thank you!

Page 5: missing dot before reference number 8 (later in the text, we found that there are sometimes missing spaces before reference numbers: eg. before reference 21)

Answer: Thank you for pointing that out. We believe that they have all been fixed now. 

In the “Initiative and Platforms” section: you should create a new paragraph for each geographic area: “In Europe”, “In Canada”, “In the US”, “In South America”, “In Asia”, “In Africa”. It will improve the clarity of this section.

Answer: New paragraphs and titles have been added. Thank you!

Page 7, in the paragraph about Canada: The reference to the OJS system (32) seems a bit out of context as the link between this system and OA is not explicated

Answer: We removed the reference to the OJS as it is not necessarily an OA focused initiative. Thank you. 

Page 10: The acronym “BOAI” is mentioned whereas it has never been introduced before

Answer: We added the acronym to the first mention of the BOAI. Thank you!

Results section, 1st sentence: Replace "Tables 1 and 2 presents" by "Tables 1 and 2 present"

Answer: Done, thank you!

Table 1 caption: missing parenthesis

Answer: Done, thank you!

Figure 5: on the left part, green is in yellow and gold is in green

Answer: Done, thank you!

Figure 1 and 2 + Appendix: the maps should be redraw with the help of the provided script (it is possible to change the colours if you wish to, and, if necessary, to add an intermediary class around zero to represent countries with z-scores equal to the world average).

Answer: We used your script to make the new figures. Thank you so much.

---

## [Editor Report · Decision Letter 2]

26 Jul 2022

National differences in dissemination and use of open access literature

PONE-D-21-29770R2

Dear Dr. Simard,

We’re pleased to inform you that your manuscript has been judged scientifically suitable for publication and will be formally accepted for publication once it meets all outstanding technical requirements.

Kind regards,

Alberto Baccini, Ph.D.

Academic Editor

PLOS ONE
---

## [Editor Report · Acceptance letter]

1 Aug 2022

PONE-D-21-29770R2 

National differences in dissemination and use of open access literature 

Dear Dr. Simard:

I'm pleased to inform you that your manuscript has been deemed suitable for publication in PLOS ONE. Congratulations! Your manuscript is now with our production department. 

Kind regards, 

on behalf of

Prof. Alberto Baccini 

Academic Editor

PLOS ONE